# Why do LLMs attend to the first token? ♻

Federico Barbero[1],[*], Álvaro Arroyo[1],[*], Xiangming Gu[2],
Christos Perivolaropoulos[3], Michael Bronstein[1], Petar Veličković[3], Razvan Pascanu[3]
[1]University of Oxford  [2]National University of Singapore  [3]Google DeepMind
[*]Equal contribution. Correspondence to federico.barbero@cs.ox.ac.uk.

## Abstract

Large Language Models (LLMs) tend to attend heavily to the first token in the sequence – creating a so-called *attention sink*. Many works have studied this phenomenon in detail, proposing various ways to either leverage or alleviate it. Attention sinks have been connected to quantisation difficulties, security issues, and streaming attention. Yet, while many works have provided conditions in which they occur or not, a critical question remains shallowly answered: *Why do LLMs learn such patterns and how are they being used?* In this work, we argue theoretically and empirically that this mechanism provides a method for LLMs to avoid over-mixing, connecting this to existing lines of work that study mathematically how information propagates in Transformers. We conduct experiments to validate our theoretical intuitions and show how choices such as context length, depth, and data packing influence the sink behaviour. We hope that this study provides a new practical perspective on why attention sinks are useful in LLMs, leading to a better understanding of the attention patterns that form during training.

## 1 Introduction

Large Language Models (LLMs) are powered by hundreds or even thousands of attention heads that are orchestrated to update the values of tokens within a sequence. As their attention patterns are the only mechanism that allows for the *mixing* of information between tokens within a Transformer architecture, it is natural to study them to understand how information is being processed. A peculiar and interesting phenomenon that has been spotted across frontier language models is that attention heads often exhibit 'attention sinks', where seemingly meaningless tokens – often the first one in the sequence – tend to capture most of the attention. Attention sinks have been linked to a number of important topics such as quantisation (Liu et al., 2024), improved KV-caching (Ge et al., 2024), streaming attention (Xiao et al., 2024), and even security vulnerabilities (Yona et al., 2025), making them an important artifact that is not yet well-understood in frontier LLMs.

While many works have studied attention sinks in order to mitigate them, in this work we take a different angle and aim to understand *why they are useful.* We do this for a simple reason: as sinks are widespread and appear as a byproduct of gradient descent rather than any explicit priors, they must provide an important mechanism for processing the context. We are therefore interested in understanding and explaining when and why the attention sink mechanism is useful. We tackle the issue from various angles: theoretically studying why such an effect is useful from a 'mixing' perspective, and performing measurements in both frontier LLMs and models we train from scratch to support our theory.

We focus specifically on understanding why attention sinks occur mostly at the first position of the context. In this position, there commonly lives a special token often denoted as ⟨bos⟩ (beginning of sequence). We find, for example, that in a typical prompt in Llama 405B, *almost 80% of the attention is concentrated on the ⟨bos⟩ token* (see Section 4.2 for details). This type of attention allocation is in many ways 'wasted', and we find it intriguing to understand exactly why this type of learned behavior is useful.

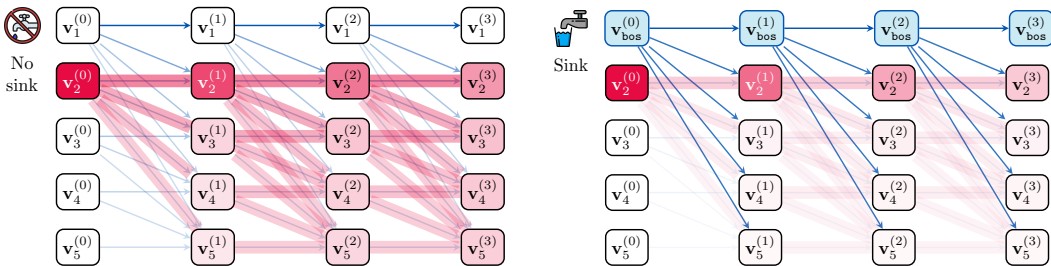

Figure 1: Our key result is to illustrate how attention sinks are usefully leveraged by decoder-only Transformers. The presence of attention sinks *slows down the mixing of information between tokens* and hence makes Transformers more robust to perturbations of prompts. To illustrate this, we demonstrate how sharply a perturbation in the second token's input representation (in **red**) affects the embeddings of other tokens throughout the model, both without (**left**) and with (**right**) a sink token (e.g. ⟨bos⟩). The presence of a sink draws attention away from the rest of the tokens, limiting the spread of perturbed information and resulting in more stable embeddings. See Figure 2 for a direct measurement in Gemma 7B.

Our main contribution is showing that this type of pattern is a way for deep Transformers to avoid *'over-mixing'*, a phenomenon related to a number of theoretical and empirical works that study rank collapse (Dong et al., 2021), representational collapse (Barbero et al., 2024), signal propagation (Noci et al., 2022; Arroyo et al., 2025), and over-smoothing (Di Giovanni et al., 2023). In particular, the depth and large context of modern LLMs seems to be sufficient to cause representational collapse, which can be slowed down by making certain heads *inactive*: a behaviour which attention sinks directly promote (see Figure 1). The deeper the architecture, the more inactive heads have to become to ensure that representations stay sufficiently separated throughout the entire architecture to avoid collapse.

**Contributions.**  We summarise our main contributions.

- In Section 3, we argue that attention sinks are useful to control over-mixing. We connect this to existing theoretical phenomena such as rank collapse, representational collapse, and over-squashing. We show how our mathematical intuitions manifest in Gemma 7B.
- In Section 4, we further support our over-mixing hypothesis. Our refined over-squashing analysis suggests that bigger models and models trained over longer contexts should have stronger sinks. We verify both hypotheses using the LLaMa 3.1 family of models and through our own pre-training runs. We find that a staggering 80% of attention heads form strong sinks in LLaMa 3.1 405B.
- In Section 5, we show that, as expected from our hypothesis, attention sinks form regardless of how ⟨bos⟩ is included during pre-training. Fixing ⟨bos⟩ in pre-training as the first token, however, does impact how the model constructs the sinks.

## 2 Background

In this work, we focus on decoder-only Transformer models (Radford et al., 2018), that apply a causal mask to the attention mechanism. These are by far the most common types of Transformer models that are used in modern LLMs to date (Gemma Team et al., 2024; Dubey et al., 2024). We follow the notation of Barbero et al. (2024), but we importantly also consider a model with $H \geq 1$ attention heads:

$$\mathbf{z}_i^{(\ell,h)} = \sum_{j \leq i} \alpha_{ij}^{(\ell,h)} \mathbf{W}^{(\ell,h)} \mathbf{v}_j^{(\ell)}, \text{ with } \alpha_{ij}^{(\ell,h)} = \frac{\exp\left(k\left(\mathbf{q}_i^{(\ell,h)}, \mathbf{k}_j^{(\ell,h)}, \mathbf{p}_{ij}\right)\right)}{\sum_{w \leq i} \exp\left(k\left(\mathbf{q}_i^{(\ell,h)}, \mathbf{k}_w^{(\ell,h)}, \mathbf{p}_{iw}\right)\right)}$$

$$\mathbf{z}_i^{(\ell)} = \mathbf{W}^{(\ell)} \bigoplus_{h \in H} \mathbf{z}_i^{(\ell,h)} + \mathbf{v}_i^{(\ell)},$$

$$\mathbf{v}_i^{(\ell+1)} = \boldsymbol{\psi}^{(\ell)}\left(\mathbf{z}_i^{(\ell)}\right) + \mathbf{z}_i^{(\ell)},$$

where $\boldsymbol{\psi}^{(\ell)}$ is a non-linearity, $k$ takes queries $\mathbf{q}$, keys $\mathbf{k}$, and positional encodings $\mathbf{p}_{ij}$ to produce an activation, $\mathbf{W}^{(\ell,h)} \in \mathbb{R}^{d \times d}$ and $\mathbf{W}^{(\ell)} \in \mathbb{R}^{Hd \times d}$ are learnable matrices, and $\oplus$ represents a direct sum (concatenation). To simplify notation, we ignore the layer normalisations without loss of generality. The sum ranging over $j$ such that $j \leq i$ is due to the causal mask. If we represent the attention coefficients via a matrix $\mathbf{A}^{(\ell,h)}$ such that $\mathbf{A}^{(\ell,h)}_{ij} = \alpha^{(\ell,h)}_{ij}$, this condition is equivalent to enforcing that $\mathbf{A}^{(\ell,h)}$ is lower triangular. An LLM consists of $L$ such blocks, with $L$ often called the *depth*. New tokens are generated autoregressively by considering the final layer representation $\mathbf{v}^{(L)}_n$ of the last token and mapping it to a distribution over the token vocabulary. A new token is sampled from this distribution, and the process repeats.

**Attention Sinks.** The term *attention sink* was first used by Xiao et al. (2024) to indicate tokens that, although likely to have limited semantic meaning, attract a large portion of the attention within an attention head. They showed that it is important to keep such tokens when computing sliding-window attention to retain performance. The attention sink phenomenon has been however observed much earlier under different names, for instance, Vig & Belinkov (2019); Vig (2019) observed the sink phenomenon calling it 'null attention', and Clark et al. (2019); Brunner et al. (2019) observed a similar 'no-op' phenomenon in the encoder-Transformer model BERT (Devlin et al., 2019).

Recent work by Gu et al. (2025) empirically ablated a number of components in the pre-training setup to study under which conditions attention sinks occur. While attention sinks are a broader term, in our work we focus on attention sinks forming exactly at the first token, as this is the most common pattern by far. To measure the presence of the sink we follow the metric proposed by Gu et al. (2025) **sink rate** $= \frac{1}{LH} \sum_{h,\ell} \mathbf{1} \left( \frac{1}{T} \sum_j \alpha^{(\ell,h)}_{1,j} > \epsilon \right)$. The quantity measures the proportion of heads in the entire model that attend to the sink on average with a coefficient of at least $\epsilon$, where we also set $\epsilon = 0.3$, unless explicitly stated.

Several key works have investigated how attention sinks are constructed. Cancedda (2024) show, from a spectral perspective, that specific subspaces are responsible for the creation of attention sinks. Sun et al. (2024) show that massive activations seem to be responsible for the creation of attention sinks. Barbero et al. (2025) reverse engineer a specific attention head to show that *high-norm bands* in the queries and keys help with the formation of attention sinks. Such works all imply that *large activations* are helpful in creating attention sinks.

**In this work**, we are interested instead in showing not only how attention sinks appear but *why they are useful*, particularly when it comes to long context modelling. In fact, we critically argue that this learned behaviour is *necessary for effective long-context learning*. We believe that that this constitutes a new perspective that nicely complements existing works.

**Propagation of information in Transformers.** Many works have studied how information propagates in (deep) Transformers. In the linear case, a phenomenon known as *rank collapse* has been heavily studied (Dong et al., 2021; Geshkovski et al., 2023; Wu et al., 2024; Naderi et al., 2024). Such works show that repeated application of attention layers projects the values into a vector space that has rank 1. The same phenomenon has been observed and heavily studied in Graph Neural Networks and is often called *over-smoothing* (Di Giovanni et al., 2023; Keriven, 2022). The key intuition is that attention matrices 'mix' information, and *repeated mixing* converges to a space that is *uninformative*. Recently, work by Wu et al. (2024) has extended such analysis to causal mechanisms and thus decoder-only Transformers, describing how causal masking affects the convergence.

Importantly, Veličković et al. (2024) proved that, when generalising to sufficiently longer contexts at inference time, global attention matrices cannot remain sharp, and will therefore always converge to "pure mixing". The culprit for this is *tokenisation*, which imposes a bound on the *logit spread* feeding into the softmax. While sparsifying attention can improve sharpness, the tradeoffs involved are not yet well understood (Vitvitskyi et al., 2025).

A related behaviour that occurs in decoder-only Transformers is *over-squashing*. Barbero et al. (2024) showed that the decoder-only Transformers are more sensitive to tokens coming

sooner in the sequence due to the causal mask. They also describe a phenomenon called *representational collapse*, in which over long sequences the Transformer tends to destroy information of tokens coming towards the end of the sequence.

Together, these effects point towards two difficulties: Transformers tend to 'over-mix' their information, both as they become deeper (Barbero et al., 2024) and as they ingest longer context (Veličković et al., 2024). **In this work**, we connect these ideas to the attention sink phenomenon. We show that the specific attention sink pattern is used by the Transformer in an attempt to counter the collapse of representations and keeping them meaningfully distant from each other. As a further contribution, we provide interesting connections between rank collapse, representational collapse, and over-squashing, which might be of independent interest.

## 3    Transformers blocks need to avoid over-mixing

We present mathematical insights that aim to understand why the formation of attention sinks can be useful or even *necessary*. We start by connecting rank and representational collapse, showing that rank collapse is a stronger condition than representational collapse. We then derive stronger over-squashing bounds and use these results to make predictions on what factors might influence the formation of attention sinks. We perform some experiments on Gemma 7B to verify our intuitions.

**Rank collapse is a stronger condition than representational collapse.**    We let $\mathbf{v}_i^{(\ell)}$ be the value vector of the $\ell$-th Transformer block of the $i$-th token and $\mathbf{V}^{(\ell)}$ the $n \times d$ matrix that collects the $n$ value vectors together. We use the same definition of rank collapse[1] from Wu et al. (2024), which in our notation can be written as:

$$\left\| \mathbf{V}^{(L)} - \frac{1}{n}\mathbf{1}\mathbf{1}^\top \mathbf{V}^{(L)} \right\|_F = \left\| \mathbf{V}^{(L)} - \hat{\mathbf{V}}^{(L)} \right\|_F < \Delta. \tag{1}$$

Rank collapse can therefore be seen as how far the representations $\mathbf{V}^{(L)}$ are from the 'average' representation $\hat{\mathbf{V}}^{(L)} = \frac{1}{n}\mathbf{1}\mathbf{1}^\top \mathbf{V}^{(L)}$. In Transformers with no residual connections and non-linearities it is well-known that this quantity decays exponentially with depth. Barbero et al. (2024) define representational collapse as[2]:

$$\left\| \mathbf{v}_n^{(L)} - \mathbf{v}_{n-1}^{(L)} \right\|_2 < \Delta \tag{2}$$

for some token sequence in which the tokens $n-1$ and $n$ are repeated and the underlying sequence or 'prefix' grows. The two measures are seemingly very related; in fact, we start by showing that rank collapse *implies* representational collapse (we provide proofs for all statements in this section in the Appendix Section D). The converse, instead, is not true, meaning that the two quantities are measuring distinct effects.

**Proposition 3.1** (Rank collapse implies representational collapse.). *If* $\|\mathbf{V}^{(L)} - \frac{1}{n}\mathbf{1}\mathbf{1}^\top \mathbf{V}^{(L)}\|_F < \Delta/2$, *then* $\|\mathbf{v}_n^{(L)} - \mathbf{v}_{n-1}^{(L)}\|_2 < \Delta$.

We highlight that this *does not* mean that representational collapse is not a useful quantity to consider. The condition of rank collapse is much stronger and only really occurs in linear systems Wu et al. (2024); Dong et al. (2021); instead, representational collapse can be studied in non-linear systems, as done by Barbero et al. (2024). Interestingly, rank collapse is a statement regarding *depth* of the model, while representational collapse is a statement

---

[1]We believe that rank collapse might not be the best name for this phenomenon as the rank can remain full for any $\Delta > 0$, but use this terminology as a matter of consistency with previous works.

[2]Note that in this work we use the $\ell^2$ norm instead of $\ell^1$ for convenience. This detail is not important as the two norms are 'equivalent' as one has the bound $\|\mathbf{x}\|_2 \leq \|\mathbf{x}\|_1 \leq \sqrt{d}\,\|\mathbf{x}\|_2$ for $\mathbf{x} \in \mathbb{R}^d$.

related to *context length* [3]. We report collapse studies in the Appendix (Section B) for the interested reader.

These phenomena are a consequence of a catastrophic *over-mixing* effect that is caused by either the depth or context length growing too much and point towards the need for a Transformer to learn *defense mechanisms* to counter such effects. The remainder of this work will then focus on exploring how the attention sink phenomenon is one such mechanism.

### 3.1 Sinks as a way to avoid over-mixing

A natural way to measure the amount of mixing is via the norm of the following Jacobian:

$$\left\| \mathcal{J}_{ij}^{(L)} \right\| = \left\| \frac{\partial \mathbf{v}_j^{(L)}}{\partial \mathbf{v}_i^{(0)}} \right\| = \left\| \sum_{k_1 \ldots k_{L-1}} \frac{\partial \mathbf{v}_j^{(L)}}{\partial \mathbf{v}_{k_{L-1}}^{(L-1)}} \cdots \frac{\partial \mathbf{v}_{k_1}^{(1)}}{\partial \mathbf{v}_i^{(0)}} \right\|. \tag{3}$$

The quantity $\left\| \mathcal{J}_{ij}^{(L)} \right\|$ measures how sensitive the token $j$'s representation at layer $L$ is to a small perturbation of token $i$. This is similar to the analysis and justification of *vanishing gradients* in recurrent models. Intuitively, Transformers should be able to *control* this quantity, or they risk running into issues such as rank collapse or representational collapse. To motivate and set the foundation for the remainder of our work, we extend the over-squashing results from Barbero et al. (2024) to now include multi-head attention. We note that just like Barbero et al. (2024), our bounds do not push the derivatives through the softmax function in order to derive a more interpretable result (see Appendix Section D for more details).

**Theorem 3.2** (More detailed over-squashing bounds.). *Let $C_{max} > 0$ be the greatest Lipschitz constant of any layer of the Transformer, $H$ be the number of heads, and $\delta_i^j$ be 1 iff $i = j$ and 0 otherwise. Let $k \in \mathcal{P}_{i \to j}$ be a path from $i$ to $j$ of length $L$. Set $\bar{\alpha}_{ij}^{(\ell)} = \sum_h \alpha_{ij}^{(\ell,h)} + \frac{\delta_i^j}{H}$. Then:*

$$\left\| \partial \mathbf{v}_j^{(L)} / \partial \mathbf{v}_i^{(0)} \right\| \leq C_{max}^L \sum_{k \in \mathcal{P}_{i \to j}} \bar{\alpha}_{j,k_{L-1}}^{(L)} \bar{\alpha}_{k_{L-1},k_{L-2}}^{(L-1)} \cdots \bar{\alpha}_{k_1,i}^{(1)}. \tag{4}$$

The bound tells us that the weighted paths *across attention heads* impact the sensitivity between tokens $i$ and $j$. From this perspective, the effect of attention sinks is clear: attention sinks help to control the impact a perturbation can have on the output, as depicted in Figure 1. Interestingly, the bound shows how the sensitivity is controlled by the depth, number of heads, and context length. We therefore expect stronger sinks to appear when the model becomes larger or is trained on longer context, to better control the sensitivity. We will provide in later sections supporting evidence on both frontier LLMs and on LMs trained from scratch.

### 3.2 How sinks help prevent mixing in Gemma 7B

To validate the intuition, we perform a perturbation analysis in Gemma 7B. To simulate a small perturbation in token space, we perturb a single token in a sequence—for example, changing 'greatest' to 'best' (see Appendix Section C for details). We then measure how the representations change across the model when the attention sink is present versus when it is not. In Figure 2 (a), we show how the perturbation behaves when ⟨bos⟩ is kept and in (b) when it is removed. It is clear that the perturbation in (b) affects the representations much more, which occurs as a consequence of the higher mixing rate. We note that this experiment is a way to estimate the behaviour of $\left\| \mathcal{J}_{ij}^{(\ell)} \right\|$. We believe the above method also acts as an interesting way of measuring over-squashing in a meaningful way, *something that was left as an open question* by Barbero et al. (2024).

---

[3]We highlight related work by Naderi et al. (2024) that studies rank collapse in 'width' through random matrix theory methods.

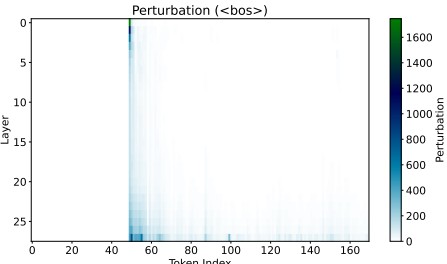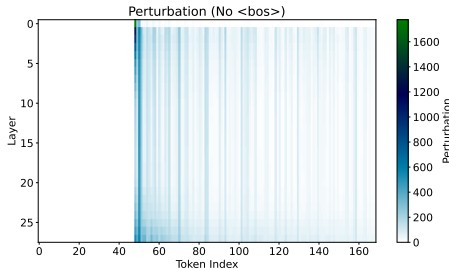

Figure 2: Effect of token perturbations on Gemma 7B. **Left/Right:** With/without ⟨bos⟩.

In Figure 3, we illustrate how the removal of the ⟨bos⟩ token in Gemma 7B causes the attention maps to become much smoother. This has the effect of increasing the values of $\|\mathcal{J}_{ij}\|$, this further supports the claim that the existence of ⟨bos⟩ provides a mechanism to attenuate the mixing, as previously shown in Figure 2.

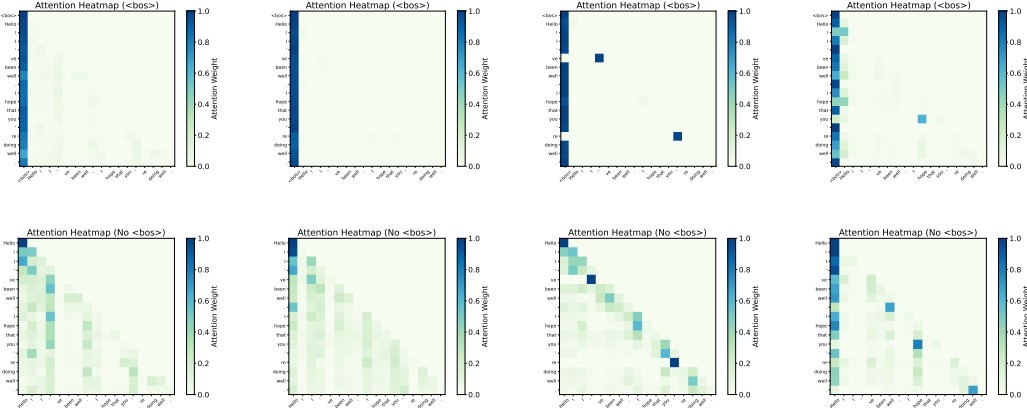

Figure 3: Attention patterns of four heads in Gemma 7B. **Top/Bottom:** With/without ⟨bos⟩.

**Attention sinks help construct approximate no-ops.** To investigate more deeply the sink phenomenon, we also examine a specific head in Gemma 7B, which was already studied in Barbero et al. (2025). This attention head pattern occurs in the first layer of the model and appears to activate specifically when there is an apostrophe token as the previous token, as shown in Figure 4 (a). The head has essentially two operating modes[4]: firing very sharply when an activation condition is met, and attending to ⟨bos⟩ otherwise. This head can therefore be seen as an implementation of an 'if-else' statement.

If we plot the corresponding value vector norms, as shown in Figure 4 (b), we see that the norm of the value corresponding to the ⟨bos⟩ token is the smallest, whereas it is *largest* for the apostrophe value – which seems to be what the head prefers to focus on. This intuitively elucidates what we believe is an interesting operating mode of attention heads: updating the token embeddings as little as possible by default[5], and instead updating them significantly when the head wishes to be operating. The attention sink in the ⟨bos⟩ token seems to provide a direct mechanism to construct this 'approximate no-op', which also been pointed out by other works (Gu et al., 2025). We also find it interesting that this head is a real-world example of the theoretically studied Bigram-Backcopy task from (Guo et al., 2024).

---

[4]At least for in-distribution examples; given Veličković et al. (2024), all of these mechanisms will eventually disperse with increasing context length.

[5]Placing a high attention coefficient on a value vector with low norm implies that the output of the attention mechanism will be low norm—and hence it will have less significance when summed with the residual stream from the previous block.

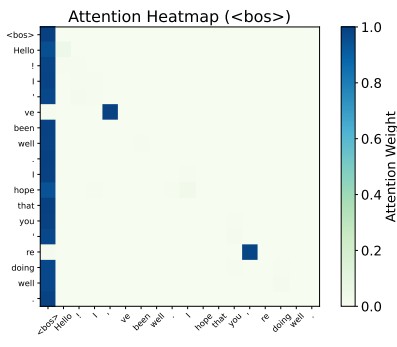
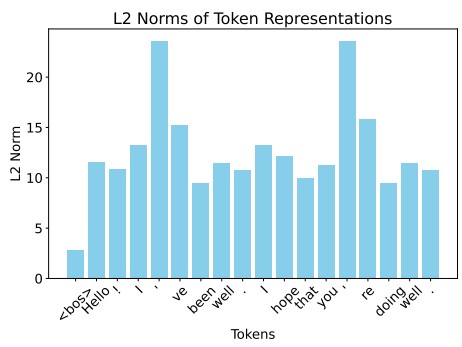

(a) Apostrophe head attention pattern

(b) Apostrophe head value norms

Figure 4: Apostrophe head attention pattern (a) and value norms (b). This attention head is a real-world example of the theoretically studied Bigram-Backcopy task from Guo et al. (2024).

> **Summary of the Section:** *Attention blocks tend to mix information, but the model needs a mechanism to control the rate of mixing to avoid pathological issues. We showed how the use of ⟨bos⟩ seems to help mitigate how perturbations spread in the computational graph.*

# 4 How does over-squashing predict attention sinks?

We now investigate how our over-squashing and mixing insights predict the formation of sinks in trained models of different sizes. Our over-squashing bounds tell us that perturbative effects will be larger in models that are trained on longer context and that are larger. Consequently, we expect this to affect sink formation if our insights are well-aligned with the underlying phenomenon.

## 4.1 How does context length affect sink formation?

While our empirical observations in pre-trained LLMs are certainly indicative of the ⟨bos⟩-powered mechanism, we are not able to meaningfully account for the way in which they were trained, or the data they observed, when reasoning about various artifacts such as the attention sink. Therefore, we follow the pre-training setup of Gu et al. (2025) and evaluate the effect of context length on the sink formation in LMs of roughly 120M parameters (see the Appendix Section A.1 for details). The over-squashing intuitions suggest that the model should learn a stronger sink as longer context naturally leads to stronger mixing (Veličković et al., 2024).

Importantly, we vary the pre-training context length, making sure that each training step processes the same amount of tokens such that the total tokens processed by each model is 5B. In Figure 5 (a) we see that, after pre-training, the attention sinks indeed seem to be much more prevalent for models trained on longer contexts, and nearly non-existent for very short-context-trained models. In Figure 5 (b), we show that this trend appears to be cumulative throughout training—initially, there are no attention sinks; the rate at which sinks develop is generally increasing with the context length (until saturating). For completeness, we also report the validation loss curves of these models in the Appendix (Figure 7), showing that they all achieve comparable validation losses during training. This is another indication that the emergence of sinks might be a *necessary* side effect of training capable models at ever-increasing context lengths in order to avoid over-mixing issues.

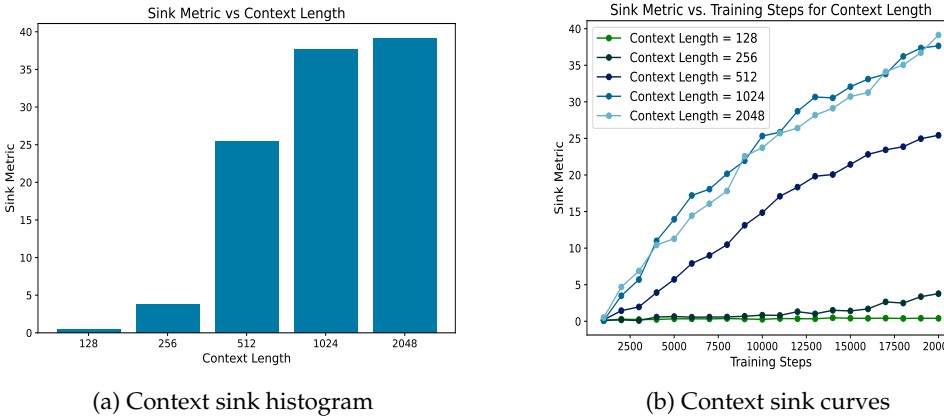

(a) Context sink histogram      (b) Context sink curves

Figure 5: Pre-trained model Sink Metric (%) after training (a) and during training (b) trained at different context lengths.

## 4.2 Attention sinks in the LLaMa 3.1 family

Next, we examine the LLaMa 3.1 family of models as it offers an interesting test bed for models of vastly different sizes. For instance, the smallest 8B model has 32 layers and 1,024 attention heads, while the largest 405B model has 126 layers and 16,128 heads (see Table 1). The underlying assumption is that as they are part of the same family of models, the training pipelines these models have undergone will be as similar as possible and allow us to study pre-trained models as their size grows. This provides an interesting way to check how attention sink patterns differ between models of different sizes.

| Model | Total Layers | Heads per Layer | Total Heads | Sink Metric ($\epsilon = 0.8$) |
|---|---|---|---|---|
| LLaMA 3.1 8B | 32 | 32 | 1,024 | 45.97 |
| LLaMA 3.1 70B | 80 | 64 | 5,120 | 73.49 |
| LLaMA 3.1 405B | 126 | 128 | 16,128 | 78.29 |

Table 1: Relevant architectural details of LLaMA 3.1 models.

In Figure 6, we show the results of our study, where we plot the sink metric for each head. We compute the metric using the same procedure and prompts as the context length experiments and sort them such that the lowest sink heads are from the left. It is evident how the smallest 8B model seems to be significantly more active than the larger models. Interestingly, the middle layers appear to be much more active, which has also been observed in other work (Skean et al., 2025). This suggests that the sink metric could also be used as a proxy of layer activity. In Table 1 we report the summary metric for each model. As the models get larger it seems like the sinks become stronger, in accordance with our intuitions from Section 3.

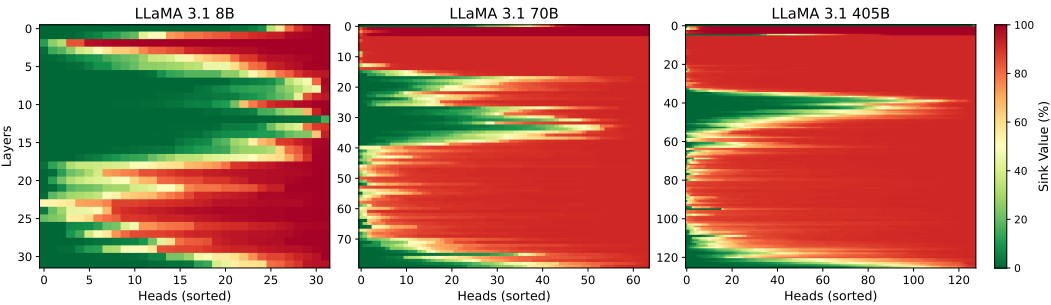

Figure 6: Sink metric percentage ($\epsilon = 0.8$) for $\langle \text{bos} \rangle$ for all heads over 170 different prompts for the LLaMa 3.1 family of models. **Red** indicates a strong sink presence.

> **Summary of the Section:** *We supported our theoretical insights that larger models and models trained on longer context should have more attention sinks to better control information-mixing.*

## 5 Is ⟨bos⟩ in any way special?

In this section we aim to answer a final important question: is there something special about the ⟨bos⟩ token and its relationship to the sink formation. Intuitively, we would expect that the answer is **no** because if sinks exist simply to prevent mixing, then the only important property the sink should have is that it exists at the first position in the sequence to help prevent mixing of the subsequent tokens.

To study this behaviour, we pre-train using a number of different strategies (see Figure 8 in the Appendix for illustrations of the different strategies). To summarise our findings from Table 2, if the model is trained with ⟨bos⟩ always appearing at the first token, removing ⟨bos⟩ during inference removes the attention sink, i.e. the model relies on the ⟨bos⟩ token to form the sink. Instead, if there is no ⟨bos⟩ during training, the sink forms at the first token regardless, but is slightly weaker. Removing ⟨bos⟩ in a model that was trained with ⟨bos⟩ present greatly reduces performance. This seems to be consistent over both causal masking and intra-doc masking. This suggests that choices in pre-training have a direct impact on how the attention sinks are constructed by the model, but that their formation during training seems inevitable. This also validates our intuition that attention sinks seem to form always at the first token, regardless of the pre-training packing strategy used. For more details, we point the reader to the Appendix (Section A.3).

| Attention Masking | ⟨bos⟩ | ⟨eos⟩ | Inference | Sink Metric (%) | Valid loss |
|---|---|---|---|---|---|
| Causal | No | Yes | ⟨bos⟩ * + text | 65.10 | 2.69 |
| Causal | No | Yes | text | 65.15 | 2.70 |
| Causal+fixed ⟨bos⟩ | Yes | Yes | ⟨bos⟩ + text | 90.84 | 2.69 |
| Causal+fixed ⟨bos⟩ | Yes | Yes | text | 0.05 | 7.56 |
| Intra-doc | No | Yes | text | 28.23 | 2.67 |
| Intra-doc | Yes | Yes | ⟨bos⟩ + text | 83.33 | 2.67 |
| Intra-doc | Yes | Yes | text | 50.24 | 2.68 |
| Intra-doc + fixed ⟨bos⟩ | Yes | Yes | ⟨bos⟩ + text | 90.56 | 2.67 |
| Intra-doc + fixed ⟨bos⟩ | Yes | Yes | text | 0.00 | 7.78 |

Table 2: Impact of data packing and attention masking on the formation of attention sink. Here * denotes that ⟨bos⟩ and ⟨eos⟩ are the same during the inference.

**Impact of attention sinks on downstream performance** For completeness, we also check how the removal of attention sinks at inference time affects performance on downstream benchmarks. We report the results in Table 3. We find that removing the attention sink (i.e., removing the ⟨bos⟩ token) in Gemma 7B at inference time consistently lowers the performance, and this drop is particularly pronounced on the long-context ruler benchmark – in accordance with our prediction that attention sinks are particularly useful for long-context. We highlight that our results suggest that one should take great care in how the ⟨bos⟩ token is handled at inference, as the model's predictive capabilities seem to be greatly impacted, especially in the long-context regime. These insights are in agreement with earlier work on streaming attention (Xiao et al., 2024) that found that the presence of attention sinks is important for long-context perplexity.

> **Summary of the Section:** *During the LM pre-training, when ⟨bos⟩ is fixed in the first position within the context, LMs employ ⟨bos⟩ to avoid over-mixing. Otherwise, LMs employ the first token (which need not be ⟨bos⟩) to avoid over-mixing.*

| Benchmark | w/ ⟨bos⟩ | w/o ⟨bos⟩ |
|---|---|---|
| ARC-e | 80.77 | 28.49 |
| ARC-c | 53.50 | 22.53 |
| PIQA | 81.72 | 52.77 |
| SIQA | 48.26 | 34.70 |
| HellaSwag | 80.61 | 27.35 |
| Winogrande | 72.85 | 49.41 |
| Ruler (4096 context) | 82.57 | 0.00 |

Table 3: Effect of removing the attention sink (i.e., the ⟨bos⟩ token) with Gemma 7B at inference time on downstream benchmarks. The performance drops significantly, even dropping to 0 on the long-context ruler task.

## 6    Conclusion

In this work, we proposed a new perspective on attention sinks, arguing that they emerge as a natural response to over-squashing and over-mixing in transformer architectures. Our analysis shows that directing a significant portion of the attention to the ⟨bos⟩ token helps the model become less sensitive to token perturbations. As models scale up in size or are trained on longer contexts, they inherently become more vulnerable to such perturbations and the sinks become stronger. We believe not only that this work helps to understand why such patterns are in fact useful, but also serves as an interesting further application of theoretical results on over-squashing and rank collapse that could be of independent interest. Our data packing exploration also helps to elucidate how the way pre-training is performed can heavily impact how attention patterns behave.

We hope that this new perspective will inspire future work aimed at deepening our understanding of the underlying mechanisms learned by transformers, ultimately guiding the development of more robust and efficient architectures.

## Acknowledgments

We would like to thank Andras Gyorgy (Google DeepMind) and Simon Osindero (Google DeepMind) for their valuable comments on our work. We are grateful to Yonatan Belinkov (Technion) for suggesting additional references.

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

## A Experimental Details and Additional Results

We provide here additional details pertaining to our experiments in the main section of the work.

### A.1 Pre-training Experimental Details

For the synthetic pre-training runs, we use the same setup as the one used by Gu et al. (2025). We train LLaMa2-style LMs with roughly 120M parameters. For the packing experiments we train on 30B tokens, while for the context length ablations we train on 5B tokens. To perform the experiments, we adapt the codebase which is released under an MIT license at https://github.com/sail-sg/Attention-Sink. This allows us to have results that are

consistent with the large number of ablations that have already been conducted by the original authors. For reference, a single training run on 5B tokens takes up to 24 hours depending on the experimental setup on a single NVIDIA H100.

## A.2 Additional Context Length Experimental Results

For completeness, we provide in Figure 7 the training loss curves for the context length ablation experiment. We find that the models trained at different context lengths achieve similar loss curve patterns. We suspect that longer training would cause the sink patterns to become sharper as this seems to be the trend as training continues. It is clear however from the plots that the context length clearly impacts very heavily the formation of the sinks, with essentially no sink formation at a context length of 128 and much stronger formation for the longer contexts.

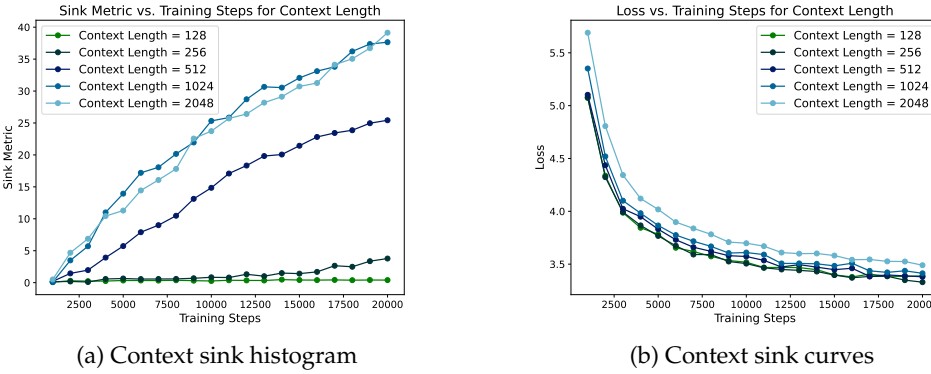

(a) Context sink histogram            (b) Context sink curves

Figure 7: Context sink curve (a) and their respective loss curves (b).

## A.3 Packing

We now provide additional intuition for our packing experiments. During LM pre-training, all documents in the corpus are concatenated and chucked into sequences with a fixed context length following Brown et al. (2020), as shown in Figure 8. Then ⟨bos⟩ (before each document) and ⟨eos⟩ (after each document) are introduced to mark the boundaries between two consecutive documents. Typically, one only needs to adopt either ⟨bos⟩ or ⟨eos⟩ during LM pre-training. In this case, ⟨bos⟩ and ⟨eos⟩ are the same during inference. To pre-train LMs, casual masking within the context is adopted, resulting in the tokens in each document being able to attend to the tokens in the previous document within the context. This motivates the proposal of intra-doc masking (Dubey et al., 2024; Zhao et al., 2024), which ensures that the tokens can only attend to previous tokens within the same document (⟨bos⟩ and ⟨eos⟩ inclusive). In this case, we consider two scenarios: (i) adding ⟨eos⟩ only, and (ii) adding both ⟨bos⟩ and ⟨eos⟩.

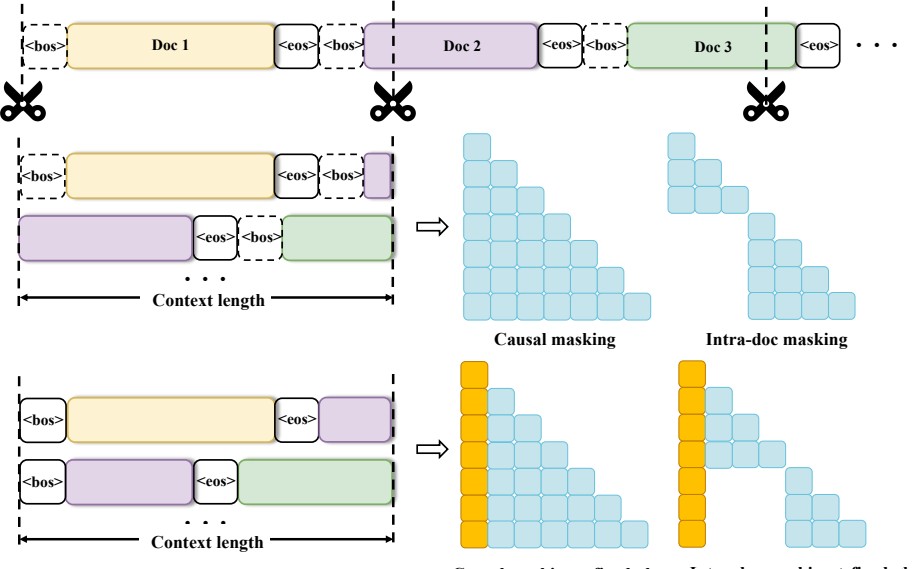

Figure 8: Illustration of data packing and attention masking setups we consider in our packing ablations in Section 5.

**Fixing ⟨bos⟩ in the first position.**    Due to the importance of the first token, we are interested in fixing ⟨bos⟩ in the first position within the context. Then only ⟨eos⟩ is added to mark the boundary between two consecutive documents. As shown in Figure 8, we modify the attention masking to ensure all tokens within the context are able to attend to the ⟨bos⟩ token regardless of intra-doc masking or not.

**The effect of ⟨bos⟩ token.**    We conduct experiments employing LLaMA2-style LMs with 120M parameters, a training corpus of 30B tokens, and a context length of 2048. We sample 100 sequences (with token lengths between 512 and 2048) in the validation set and measure the auto-regressive loss and attention sink metric for the first token position following Gu et al. (2025). As shown in Table 2, we first consider the scenarios of causal masking, if ⟨bos⟩ is only used as the boundary for different documents, it has little impact on sink formation. Meanwhile, if we fix ⟨bos⟩ in the first position within the context, the attention sink is significantly enhanced. But without ⟨bos⟩, the attention sink disappears, and the model performance drops drastically. When it comes to intra-doc masking, adopting ⟨bos⟩ in the inference significantly improves attention sink. But without it, LMs still have attention sink on the first token. If we fix ⟨bos⟩ in the first position during pre-training, removing it during inference will make LMs experience no attention sink and a drop in performance. To summarize, when ⟨bos⟩ is fixed in the first position within context during the pre-training, LMs employ ⟨bos⟩ to avoid over-mixing. Otherwise, LMs employ the first token (which need not be ⟨bos⟩) to avoid over-mixing.

# B  How does BoS help avoid over-mixing?

At each transformer layer, the learned attention matrix mixes different tokens which are then fed into the MLPs to construct intermediate representations. While mixing is crucial to capture meaningful semantics in sentences, the spectral characteristics of attention matrices with causal masking leads to the *collapse* of token representations. This phenomenon has been termed *rank collapse* (Dong et al., 2021), and is closely related to the *oversmoothing* phenomenon in GNNs (Di Giovanni et al., 2023) and signal propagation (Noci et al., 2022; Arroyo et al., 2025).

Here, we quantify token similarity by measuring the deviation of the representations from their mean, using the distance measure presented in (1), which we denote $\mu(\mathbf{X})$. $\mu$

measures how far **X** is from its 'mean matrix'. As an initial empirical test, we measure the representational distance to the mean at the final layer of the Gemma 7B model for progressively longer prompts asking the model to add numbers. We provide more details on these prompts in Appendix C. The results are shown in Figure 9 below.

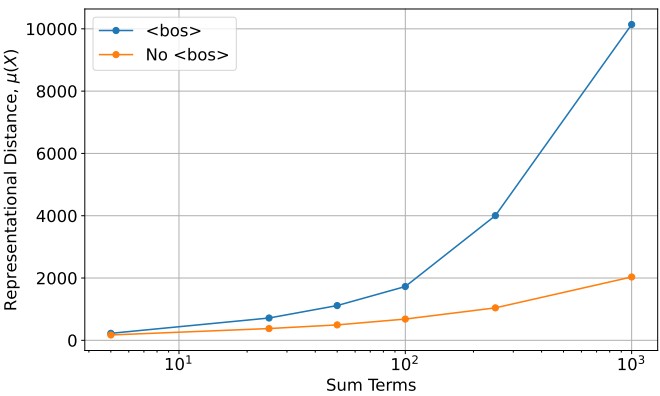

Figure 9: Representational distance as a function of prompt length for models with an without BoS token for Gemma 7B.

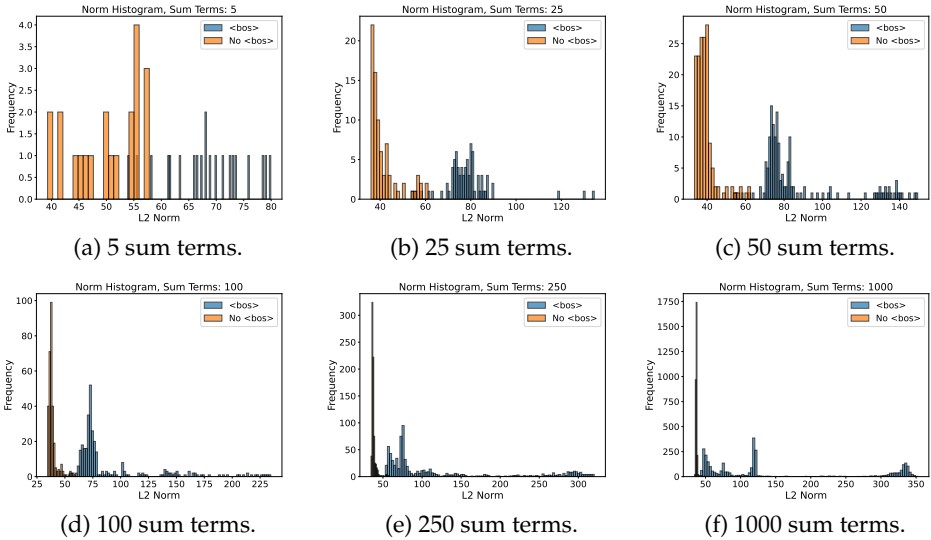

(a) 5 sum terms.   (b) 25 sum terms.   (c) 50 sum terms.

(d) 100 sum terms.   (e) 250 sum terms.   (f) 1000 sum terms.

Figure 10: Histogram of token norms for sums of increasing length.

Figure 4 illustrates that excluding the BoS token when prompting an LLM significantly affects the smoothness of its representations. Specifically, as the prompt length increases, the divergence between representations with and without the BoS token grows. Notably, prompts that include the BoS token exhibit considerably higher dispersion around their mean. This dispersion can also be visualized by plotting the histogram of the norms of the latent representation of each token, as shown in Figure 10.

The empirical evidence in Figure 4 serves to clarify the effect of the ⟨bos⟩ token on the formation of intermediate representations. The disproportionally low norm of the values associated with the BoS token implies that if the model directs all attention to that token, the resultant information is essentially nullified — effectively bypassing the mixing process intrinsic to attention. In such scenarios, only the residual pathway carries forward any

information. This process counteracts the over-mixing and rank collapse induced by the spectral properties of the attention matrix with causal masking (Naderi et al., 2024). Notably, this strategy mirrors the mixture-of-depths approach introduced in Raposo et al. (2024), but without the efficiency gains achieved through gating around the attention operation.

## C   Additional Details on Prompts Used

We include additional details on the prompts used for the experiments presented in the paper.

### C.1   Robustness to Perturbations Prompts

Here, we present the prompts used to obtain the results in Figure 2. The original prompt is

```
William Shakespeare[a] (c. 23[b] April 1564, 23 April 1616)[c] was an English
playwright, poet and actor. He is widely regarded as the greatest writer in
the English language and the world's pre-eminent dramatist. He is often called
England's national poet and the 'Bard of Avon' (or simply 'the Bard'). His extant
works, including collaborations, consist of some 39 plays, 154 sonnets, three
long narrative poems and a few other verses, some of uncertain authorship. His
plays have been translated into every major living language and are performed
more often than those of any other playwright. Shakespeare remains arguably the
most influential writer in the English language, and his works continue to be
studied and reinterpreted.
```

while the perturbed prompt is:

```
William Shakespeare[a] (c. 23[b] April 1564, 23 April 1616)[c] was an English
playwright, poet and actor. He is widely regarded as the best writer in the
English language and the world's pre-eminent dramatist. He is often called
England's national poet and the 'Bard of Avon' (or simply 'the Bard'). His extant
works, including collaborations, consist of some 39 plays, 154 sonnets, three
long narrative poems and a few other verses, some of uncertain authorship. His
plays have been translated into every major living language and are performed
more often than those of any other playwright. Shakespeare remains arguably the
most influential writer in the English language, and his works continue to be
studied and reinterpreted.
```

Here, the perturbed word is written in purple. The modification critically keeps the number of tokens constant and is chosen specifically to be a 'small' perturbation.

### C.2   Smoothing Effects Prompts

To empirically investigate the smoothing effects in transformer representations, we designed a set of prompts that probe the model's ability to maintain distinct token representations over long contexts. In these prompts, the model is asked to calculate a sum of numbers that gets progressively longer (denoted with Sum Terms). In our experiments, we compare the effects of including versus omitting the ⟨bos⟩ token on the smoothness of the resulting representations. One representative example of these prompts is:

- ```Could you add these numbers 10+06+03+10+08```

### C.3   LLaMa Sink Prompts

For the LLaMa sink prompts, we use the dataset provided by Gu et al. (2025) which is found at https://github.com/sail-sg/Attention-Sink/blob/main/datasets/probe_valid.jsonl. We evaluate on all of the 170 samples and measure the sink metric on the first $T = 64$ tokens to remain consistent with Gu et al. (2025). Below we report a few excerpts:

**(i)** ```Role of the immune system in cardiac tissue damage and repair following myocardial infarction.\nThe immune system plays a crucial role in the initiation,```

development, and resolution of inflammation following myocardial infarction (MI). The lack of oxygen and nutrients causes the death of cardiomyocytes and leads to the exposure of danger-associated molecular patterns that are recognized by the immune system to initiate inflammation. At the initial stage of post-MI inflammation, [...]

**(ii)** `request_number: PS-BPA 116 and 118\nfirstname : Steve\nlastname : Marshall\ne` `-mail: marss@perkinscoie.com\ndirected _to: Peter J. Burger - LP-7\nOffice of` `General Counsel\nBonneville Power Administration\nP .O. Box 3261\nPortland ,` `Oregon 97208-3621\nexhibit _wp-02-e-: BPA-78\npage _numbers: 1-8\nrequest _text:` `[...]`

**(iii)** `<A     HREF=\"http://finmath.com/\">FinMath.com     @     Chicago\nFinancial` `Engineering     &     Risk     Management     Workshop</A>\n         \n` `--------------------------------------------------------------------------\n` `NEW 60-65\% OFF 2001 subscription for RISK Magazine\nfor members of Bachelier` `Finance Society.\nBecome a Member NOW! [...]`

**(iv)** `ubuntu-au 2011-10-24\n <ikt> Hi all :)\n <gggs> hi ikt \n <ikt> gggs: whatcha` `up to?\n <gggs> right now?\n <gggs> talking to you!\n <ikt> :D\n <ikt> thinking` `of setting up an LDAP server at home\n <ikt> [...]`

The 170 texts span a number of different domains, which we believe is important to appropriately evaluate the sink phenomenon.

## D  Mathematical Results

We provide here the details of the mathematical results in our paper. We start by showing the connection between representational and rank collapse.

**Proposition 3.1** *If $\|\mathbf{V}^{(L)} - \frac{1}{n}\mathbf{1}\mathbf{1}^\top\mathbf{V}^{(L)}\|_F < \Delta/2$, then $\|\mathbf{v}_n^{(L)} - \mathbf{v}_{n-1}^{(L)}\|_2 < \Delta$.*

*Proof.* Assume that we have rank collapse at $\Delta/2$:

$$\left\|\mathbf{V}^{(L)} - \frac{1}{n}\mathbf{1}\mathbf{1}^\top\mathbf{V}^{(L)}\right\|_F = \left\|\mathrm{vec}\left(\mathbf{V}^{(L)} - \frac{1}{n}\mathbf{1}\mathbf{1}^\top\mathbf{V}^{(L)}\right)\right\|_2 < \Delta/2.$$

We now bound the representational collapse quantity:

$$\begin{aligned}\left\|\mathbf{v}_n^{(L)} - \mathbf{v}_{n-1}^{(L)}\right\|_2 &\leq \left\|\mathbf{v}_n^{(L)} - \frac{1}{n}\mathbf{1}^\top\mathbf{V}^{(L)}\right\|_2 + \left\|\mathbf{v}_{n-1}^{(L)} - \frac{1}{n}\mathbf{1}^\top\mathbf{V}^{(L)}\right\|_2 \\ &\leq \left\|\mathrm{vec}\left(\mathbf{V}^{(L)} - \frac{1}{n}\mathbf{1}\mathbf{1}^\top\mathbf{V}^{(L)}\right)\right\|_2 + \left\|\mathrm{vec}\left(\mathbf{V}^{(L)} - \frac{1}{n}\mathbf{1}\mathbf{1}^\top\mathbf{V}^{(L)}\right)\right\|_2 \\ &= 2\left\|\mathbf{V}^{(L)} - \frac{1}{n}\mathbf{1}\mathbf{1}^\top\mathbf{V}^{(L)}\right\|_F \\ &< \Delta.\end{aligned}$$

Thus, rank collapse at $\Delta/2$ implies representational collapse of at least $\Delta$.  □

We now show the results on over-squashing that now include multi-head attention. Like Barbero et al. (2024), we assume for simplicity that the queries and keys are independent of the values. This is to not distract from the main point of the result as it would otherwise make the result very messy due to the need to push the partial derivatives through the softmax function. For convenience, we rewrite here the update equations that we consider:

$$\mathbf{z}_i^{(\ell,h)} = \sum_{j \leq i} \alpha_{ij}^{(\ell,h)} \mathbf{W}^{(\ell,h)} \mathbf{v}_j^{(\ell)},$$

$$\mathbf{z}_i^{(\ell)} = \mathbf{W}^{(\ell)} \bigoplus_{h \in H} \mathbf{z}_i^{(\ell,h)} + \mathbf{v}_i^{(\ell)},$$

$$\mathbf{v}_i^{(\ell+1)} = \boldsymbol{\psi}^{(\ell)} \left( \mathbf{z}_i^{(\ell)} \right) + \mathbf{z}_i^{(\ell)}.$$

For the dimensions, we let $\mathbf{v}_j^{(\ell)} \in \mathbb{R}^d$, $\mathbf{W}^{(\ell,h)} \in \mathbb{R}^{d \times d}$, and $\mathbf{W}^{(\ell)} \in \mathbb{R}^{dH \times d}$. This means that the partials $\dfrac{\partial \mathbf{v}_j^{(\ell)}}{\partial \mathbf{v}_i^{(\ell-1)}}$ are in $\mathbb{R}^{d \times d}$. Our result applies for matrix norms that are sub-multiplicative. This is true for induced operator norms, which include most common norms used traditionally in machine learning such as $\ell^1$ or $\ell^2$ vector norms and their corresponding matrix norms.

**Theorem 3.2** *Let $C_{max} > 0$ be the greatest Lipschitz constant of any layer of a Transformer, $H$ be the number of heads, and $\delta_i^j$ be 1 iff $i = j$ and 0 otherwise. Let $k \in \mathcal{P}_{i \to j}$ be a path from $i$ to $j$ of length $L$. Set $\bar{\alpha}_{ij}^{(\ell)} = \sum_h \alpha_{ij}^{(\ell,h)} + \frac{\delta_i^j}{H}$. Then:*

$$\left\| \partial \mathbf{v}_j^{(L)} / \partial \mathbf{v}_i^{(0)} \right\| \leq C_{max}^L \sum_{k \in \mathcal{P}_{i \to j}} \bar{\alpha}_{j,k_{L-1}}^{(L)} \bar{\alpha}_{k_{L-1},k_{L-2}}^{(L-1)} \cdots \bar{\alpha}_{k_1,i}^{(1)}. \tag{5}$$

*Proof.* We start by decomposing the target partial derivative:

$$\left\| \mathcal{J}_{ij}^{(L)} \right\| = \left\| \frac{\partial \mathbf{v}_j^{(L)}}{\partial \mathbf{v}_i^{(0)}} \right\| = \left\| \sum_{k_{L-1}} \frac{\partial \mathbf{v}_j^{(L)}}{\partial \mathbf{v}_{k_{L-1}}^{(L-1)}} \frac{\partial \mathbf{v}_{k_{L-1}}^{(L-1)}}{\partial \mathbf{v}_i^{(0)}} \right\| = \cdots = \left\| \sum_{k_{L-1},\ldots,k_1} \frac{\partial \mathbf{v}_j^{(L)}}{\partial \mathbf{v}_{k_{L-1}}^{(L-1)}} \cdots \frac{\partial \mathbf{v}_{k_1}^{(1)}}{\partial \mathbf{v}_i^{(0)}} \right\|.$$

We now focus on bounding the norm of the key sub-partial which is $\partial \mathbf{v}_j^{(\ell)} / \partial \mathbf{v}_i^{(\ell-1)}$. We note that the computational graph looks like $\mathbf{v}_i^{(\ell-1)} \to \mathbf{z}_i^{(\ell-1,h)} \to \mathbf{z}_i^{(\ell-1)} \to \mathbf{v}_i^{(\ell)}$. We decompose the partial of interest:

$$\left\| \frac{\partial \mathbf{v}_j^{(\ell)}}{\partial \mathbf{v}_i^{(\ell-1)}} \right\| = \left\| \sum_k \frac{\partial \mathbf{v}_j^{(\ell)}}{\partial \mathbf{z}_k^{(\ell-1)}} \frac{\partial \mathbf{z}_k^{(\ell-1)}}{\partial \mathbf{v}_i^{(\ell-1)}} \right\| = \left\| \frac{\partial \mathbf{v}_j^{(\ell)}}{\partial \mathbf{z}_j^{(\ell-1)}} \frac{\partial \mathbf{z}_j^{(\ell-1)}}{\partial \mathbf{v}_i^{(\ell-1)}} \right\|.$$

We now bound the two components, where we get:

$$\left\| \frac{\partial \mathbf{v}_j^{(\ell)}}{\partial \mathbf{z}_j^{(\ell-1)}} \right\| = \left\| \frac{\partial}{\partial \mathbf{z}_j^{(\ell-1)}} \left[ \boldsymbol{\psi}^{(\ell)} \left( \mathbf{z}_j^{(\ell-1)} \right) + \mathbf{z}_j^{(\ell-1)} \right] \right\| \leq \|\boldsymbol{\psi}\| + 1,$$

and

$$
\begin{aligned}
\left\| \frac{\partial \mathbf{z}_j^{(\ell-1)}}{\partial \mathbf{v}_i^{(\ell-1)}} \right\| &= \left\| \frac{\partial}{\partial \mathbf{v}_i^{(\ell-1)}} \left[ \mathbf{W}^{(\ell-1)} \left( \bigoplus_{h \in H} \mathbf{z}_j^{(\ell-1,h)} \right) + \mathbf{v}_j^{(\ell-1)} \right] \right\| \\
&= \left\| \frac{\partial}{\partial \mathbf{v}_i^{(\ell-1)}} \left[ \mathbf{W}^{(\ell-1)} \left( \bigoplus_{h \in H} \sum_{k \leq j} \alpha_{jk}^{(\ell-1,h)} \mathbf{W}^{(\ell-1,h)} \mathbf{v}_k^{(\ell-1)} \right) + \mathbf{v}_j^{(\ell-1)} \right] \right\| \\
&= \left\| \mathbf{W}^{(\ell-1)} \frac{\partial}{\partial \mathbf{v}_i^{(\ell-1)}} \left( \bigoplus_{h \in H} \sum_{k \leq j} \alpha_{jk}^{(\ell-1,h)} \mathbf{W}^{(\ell-1,h)} \mathbf{v}_k^{(\ell-1)} \right) + \frac{\partial}{\partial \mathbf{v}_i^{(\ell-1)}} \mathbf{v}_j^{(\ell-1)} \right\| \\
&= \left\| \mathbf{W}^{(\ell-1)} \left( \bigoplus_{h \in H} \frac{\partial}{\partial \mathbf{v}_i^{(\ell-1)}} \sum_{k \leq j} \alpha_{jk}^{(\ell-1,h)} \mathbf{W}^{(\ell-1,h)} \mathbf{v}_k^{(\ell-1)} \right) + \mathbf{I}_{d \times d} \delta_i^j \right\| \\
&= \left\| \mathbf{W}^{(\ell-1)} \left( \bigoplus_{h \in H} \alpha_{ji}^{(\ell-1,h)} \mathbf{W}^{(\ell-1,h)} \mathbf{I}_{d \times d} \right) + \mathbf{I}_{d \times d} \delta_i^j \right\| \\
&\leq \left\| \mathbf{W}^{(\ell-1)} \left( \bigoplus_{h \in H} \alpha_{ji}^{(\ell-1,h)} \mathbf{W}^{(\ell-1,h)} \mathbf{I}_{d \times d} \right) \right\| + \left\| \mathbf{I}_{d \times d} \delta_i^j \right\| \\
&\leq \left\| \mathbf{W}^{(\ell-1)} \right\| \sum_{h \in H} \left\| \mathbf{W}^{(\ell-1,h)} \right\| \alpha_{ji}^{(\ell-1,h)} + \delta_i^j \\
&\leq \left\| \mathbf{W}^{(\ell-1)} \right\| \left\| \mathbf{W}^{(\ell-1,h_{max})} \right\| \sum_{h \in H} \alpha_{ji}^{(\ell-1,h)} + \delta_i^j.
\end{aligned}
$$

This finally gives us the upper bound:

$$
\begin{aligned}
\left\| \frac{\partial \mathbf{v}_j^{(\ell)}}{\partial \mathbf{v}_i^{(\ell-1)}} \right\| &\leq (\|\boldsymbol{\psi}\| + 1) \left( \left\| \mathbf{W}^{(\ell-1)} \right\| \left\| \mathbf{W}^{(\ell-1,h_{max})} \right\| \sum_{h \in H} \alpha_{ji}^{(\ell-1,h)} + \delta_i^j \right) \\
&= (\|\boldsymbol{\psi}\| + 1) \left( \left\| \mathbf{W}^{(\ell-1)} \right\| \left\| \mathbf{W}^{(\ell-1,h_{max})} \right\| \sum_{h \in H} \left( \alpha_{ji}^{(\ell-1,h)} + \frac{\delta_i^j}{H} \right) \right) \\
&= C_\ell \sum_{h \in H} \left( \alpha_{ji}^{(\ell-1,h)} + \frac{\delta_i^j}{H} \right),
\end{aligned}
$$

where we let $C_\ell$ be a function of the Lipschiztness of the $\ell$-th Transformer block. Putting this all together we reach the desired result:

$$
\begin{aligned}
\left\| \frac{\partial \mathbf{v}_j^{(L)}}{\partial \mathbf{v}_i^{(0)}} \right\| &= \left\| \sum_{k_{L-1},\ldots,k_1} \frac{\partial \mathbf{v}_j^{(L)}}{\partial \mathbf{v}_{k_{L-1}}^{(L-1)}} \cdots \frac{\partial \mathbf{v}_{k_1}^{(1)}}{\partial \mathbf{v}_i^{(0)}} \right\| \\
&\leq \sum_{k_{L-1},\ldots,k_1} \prod_\ell \left\| \frac{\partial \mathbf{v}_j^{(\ell)}}{\partial \mathbf{v}_{k_{\ell-1}}^{(\ell-1)}} \right\| \\
&\leq C_{max}^L \sum_{k_{L-1},\ldots,k_1} \prod_\ell \sum_{h \in H} \left( \alpha_{ji}^{(\ell-1,h)} + \frac{\delta_i^j}{H} \right) \\
&= C_{max}^L \sum_{k \in \mathcal{P}_{i \to j}} \bar{\alpha}_{j,k_{L-1}}^{(L)} \bar{\alpha}_{k_{L-1},k_{L-2}}^{(L-1)} \cdots \bar{\alpha}_{k_1,i}^{(1)}.
\end{aligned}
$$

Above, $k \in \mathcal{P}_{i \to j}$ is a walk of length $L$ from $i$ to $j$ with steps $k_1, \ldots k_{L-1}, k_L = j$ that respects the causal attention structure.

□

