# OpenReview forum: "Why do LLMs attend to the first token?"
_colmweb.org/COLM/2025/Conference — COLM 2025_

### Official Review · Reviewer_Maou · 2025-05-09

**Rating:** 9
**Confidence:** 3
**Ethics Flag:** 1

**Summary:**

In this paper, the authors explore the tendency of LLMs to attribute a substantial amount of attention to the first token of the sequence.

The authors start by connecting this phenomenon with rank collapse, representation collapse, and over-squashing.
Moreover, they extend the over-squashing bounds to include multi-head attention, showing that weighted paths across attention heads impact the sensitivity between tokens which in turns makes it clear that attention sinks help controlling the impact a perturbation can cause in the output.
This bound also shows that the sensitivity is controlled by the model’s depth, number of attention heads, and context length, giving the intuition than attention sinks should be stronger for larger models or trained with longer contexts.

To corroborate the claim that attention sinks help controlling the impact a perturbation, the authors observe what happens when removing the <bos> token from the context when using Gemma 7B. This experiment shows that this removal causes the attention maps to become smoother which supports the claim.

Then, the authors also perform experiments to check the influence of the model’s depth and context length (as shown by the over-squashing bounds).
First, the authors train language models with different context lengths. The results clearly show that models trained with larger context lengths lead to stronger attention sinks and that these get stronger along the training process, indicating that the model learns to attribute attention to the first token in order to avoid over-mixing.
Then, the authors measure the sink metric percentage for models of the LLaMa-3.1 family of different sizes. Results show that the bigger models have stronger attention sinks.
These two experiments support the theoretical insights provided by the over-squashing bound.

Finally, the authors analyze the effect of the <bos> token, concluding that removing the <bos> token during inference removes the attention sink, considerably hurting the model performance and that models trained without the <bos> token, have the sink at the first token.

**Reasons To Accept:**

-	The paper is very clear and well written.
-	The paper is very well structured. The authors start the paper by providing a theoretical perspective of the attention sink phenomenon and connecting it with rank collapse, representation collapse, and over-squashing, arguing that they emerge to control the over-mixing. This is then supported by the thorough experiments.
-	The paper provides another perspective on a very noticed and not well-understood phenomenon that can lead to a deeper understanding of the transformer architecture and guide the development of more efficient architectures.

**Reasons To Reject:**

-	The only aspect that I believe the paper could be missing is a discussion (or even some small experiments) on how these findings can lead to the development of more efficient architectures, but it’s also totally understandable to leave this to future work.

---

> ### Author Response · Authors · 2025-06-01
>
> We thank you very much for your thorough review. We are happy to see that you value our work!
>
> We agree that a more detailed discussion would help guide future work. We have added this in the Appendix (with a pointer in the main text). In fact, we are very happy to see that our work has already been used by others to propose principled mitigations, which we unfortunately cannot cite to avoid breaking double blind. We will mention these follow up works in the section as well.

---

### Official Review · Reviewer_Z1TA · 2025-05-11

**Rating:** 7
**Confidence:** 4
**Ethics Flag:** 1

**Summary:**

This paper investigates the phenomenon of attention sinks in LLMs, where the first token (often 〈bos〉) attracts disproportionately high attention weights. The authors argue that this behavior is a learned mechanism to mitigate over-mixing (rank collapse, representational collapse) in deep Transformers, especially in long-context scenarios. Through theoretical analysis (connecting attention sinks to Jacobian norms and over-squashing) and empirical validation (perturbation studies, context-length ablations, and model-size comparisons), the paper demonstrates that attention sinks act as "anchors" to stabilize information propagation. The work also explores how pre-training strategies (e.g., 〈bos〉 placement) influence sink formation.

**Questions To Authors:**

1) Attention Sink as a Learned Shortcut? The observed attention sink phenomenon might simply reflect models learning a computational "shortcut." Given that >99% of next-token predictions likely depend only on local context, models could be exploiting this redundancy to minimize unnecessary attention computation. However, this raises a paradox: KV caching still requires storing all historical states despite their sparse utilization. The paper's claim that this "store-but-ignore" behavior benefits model performance lacks convincing empirical validation.

2) What fundamentally drives the consistent emergence of attention sinks at the first token (e.g., 〈bos〉) rather than other positions?  Is this driven by positional encodings, causal masking, or optimization dynamics?

3) Could sinks be replaced by explicit architectural interventions (e.g., sparse attention, gating mechanisms, or RNN layers)

4) Do attention sinks hinder performance on tasks requiring long-range dependencies (e.g., coreference resolution)?

**Reasons To Accept:**

1) Novel Theoretical Insight: The paper provides a compelling explanation for attention sinks by linking them to over-mixing avoidance, a perspective not thoroughly explored in prior work. The connection to rank collapse (Proposition 3.1) and multi-head over-squashing bounds (Theorem 3.2) is rigorous and insightful. The idea that sinks serve as "approximate no-ops" (Section 3.3) to limit unnecessary mixing is particularly elegant.

2) Strong Empirical results: Experiments on Gemma 7B and LLaMA 3.1 models convincingly validate the theory that: First, [erturbation studies (Figures 2–3) show that sinks reduce cross-token sensitivity. Second, Context-length and model-size experiments (Figures 5–6) confirm that sinks emerge more strongly in deeper/longer-context models. The ablation of pre-training strategies (Table 2) adds practical relevance for LLM training.

3) Broader Implications: The work bridges theoretical phenomena (rank collapse, over-squashing) and empirical observations (attention patterns), offering a unified framework.

**Reasons To Reject:**

1) While this paper theoretically explains why "LLMs attend to the first token," it remains unclear how this conclusion can practically enhance specific LLM capabilities (e.g., long-context processing, complex reasoning, or instruction following) or address existing limitations. The study would benefit from explicitly connecting its findings to actionable improvements in model design or training.

2) The paper argues that sinks are beneficial but overlooks potential drawbacks. For instance:

     Could heavy attention allocation to 〈bos〉 (e.g., 80% weights) impair the model’s ability to capture legitimate long-range dependencies?

     Does this phenomenon introduce inefficiencies in tasks requiring fine-grained attention to distant tokens?

---

> ### Author Response · Authors · 2025-06-01
>
> We thank you very much for having thoroughly read our work and we are happy to see that you view it in a positive light, having “novel theoretical insights”, “strong empirical results”, and “offering a unified framework”. We answer your questions below:
>
> > (Q) Attention sink as a learned shortcut
>
> We thank you for bringing up this point. We believe that the attention sink phenomenon (or, as you refer to it, “store-and-ignore”) is quite crucial for the performance of the model. This can be for example seen in Table 2, where we show that removing the BoS token has extremely detrimental effects on performance. We have also added additional ablations using real-world benchmarks (please see the new tables shared with Reviewer x9S1), which helps to further support the claim that indeed the sink phenomenon is playing an important role. For instance, we find that performance drops to 0% on the Ruler (4096 token, long range benchmark) task if one removes the attention sink. We believe there is ample evidence (from us and others) that attention sinks are particularly important for long-range tasks, which is nicely aligned with the over-mixing intuition as mixing becomes more and more problematic as the context size increases.
>
> You make a great point with regard to many operations becoming unnecessary due to the appearance of attention sinks. We have therefore added in the appendix additional comments related to the efficiency implications of our results, which are also in line with the suggestions of Reviewer Maou.
>
> > (Q) What drives the sink on the first token
>
> The work of Gu et al [1] (Table 3) shows that attention sinks emerge very consistently regardless of positional encodings used (even if no positional encodings are used, i.e. NoPE), so we can likely exclude this as the main cause. It has also been shown that attention sinks occur in encoder models as well, such as BERT [2] and even in VITs [3], which do not use a causal mask. As such, we would like to claim that such a mechanism to prevent mixing is necessary on a variety of domains and independent of the causal mask, as long as the Transformer is trained sufficiently and over a large enough context (as shown in our experiments).
>
> As for decoder-Transformers, we believe that our Section 5 helps to clarify that even ablating different pre-training packings and maskings the first token seems to be consistently chosen – this is consistent with other findings from Gu et al [1]. We believe the casual mask is largely at play here in this case. The first token is always present in the computation and therefore provides a useful mechanism to consistently prevent over-mixing – a sink formed on any other token would not always be present. In non-causal Transformers, such as fully-connected VITs, the Transformer instead uses background “useless” pixels, as shown by Darcet et al [3].
>
> > (Q) Could sinks be replaced by architectural interventions
>
> This is a great question and we believe the answer is a definite yes. Not only has this already been shown by Gu et al [1], but we are also very happy to see that several recent works have founds ways to prevent overmixing from arising through several model improvements.  We highlight the work of Zuhri et. al. [6], which introduces a variant of the softmax function with the goal of preventing attention sinks, as well as the recent work by the Qwen team [7], which introduces gating into the attention mechanism to prevent the appearance of attention sinks. We will explicitly mention some of these works in light of their importance and relevance as these techniques (such as gating), can be seen as a way for the model to “skip” the mixing effects of the attention mechanism in a more principled manner.

---

> > ### Author Response · Authors · 2025-06-01
> >
> > > (Q) Do attention sinks hinder performance on tasks requiring long-range dependencies.
> >
> > This is a very interesting question. Our view is that attention sinks are extremely useful in long-range tasks over a long context. We would like to support this using different pieces of evidence. We find in Figure 5 that attention sinks form most rapidly when training with larger context. Works such as the popular one by Xiao et al [4] show that performance drops most drastically in long contexts when attention sinks are removed (Figure 3). From a more mechanical perspective, attention sinks help to directly mitigate over-mixing effects that naturally arise due to long-context work (e.g. the issues mentioned in Veličković et al [5]).
> >
> > Given these pieces of evidence, we believe that attention sinks do not in fact harm performance, but help to prevent the natural mixing tendency that arises due to the attention mechanism that would quickly become problematic. Put differently, if in a standard LLM architecture no attention sink forms, then over long-context attention *has* to over-mix. This can be seen directly mathematically from for example mathematical results in [5], e.g. Theorem 2.2.
> >
> > Additionally, we have run an ablation on the ruler Benchmark with Gemma 7B – a task that is designed to evaluate long-range capacity. We report the results below, showing that removing the attention sink causes Gemma’s performance to drop to 0%. We believe this further supports our claim that attention sinks are particularly important for long-context as over-mixing becomes more problematic as the context length increases.
> >
> >
> > | Benchmark            | w/ <BOS> | w/o <BOS> |
> > |----------------------|----------|-----------|
> > | Ruler 4096 Context   |  82.57   |   0.00    |
> >
> >
> > We thank you again for your great questions and your careful review. We are finally very glad that you view this paper in a positive light and hope to have further strengthened your opinion of our work.
> >
> > [1] Xiangming Gu, et al. When attention sink emerges in language models: An empirical view. In The Thirteenth International Conference on Learning Representations, 2025
> >
> > [2] Kevin Clark, et al, Proceedings of the 2019 ACL Workshop BlackboxNLP: Analyzing and Interpreting Neural Networks for NLP, pp. 276–286, Florence, Italy, August 2019. Association for Computational Linguistics.
> >
> > [3] Timothee Darcet , et al. Vision Transformers need registers. In The Twelfth International Conference on Learning Representations, 2024
> >
> > [4] Guangxuan Xiao, et al. Efficient streaming language models with attention sinks. In The Twelfth International Conference on Learning Representations, 2024
> >
> > [5] Veličković, Petar, et al. "softmax is not enough (for sharp out-of-distribution)." ICML 2025.
> >
> > [6] Zuhri, Zayd MK, Erland Hilman Fuadi, and Alham Fikri Aji. "Softpick: No Attention Sink, No Massive Activations with Rectified Softmax." arXiv preprint arXiv:2504.20966 (2025).
> >
> > [7] Qiu, Zihan, et al. "Gated Attention for Large Language Models: Non-linearity, Sparsity, and Attention-Sink-Free." arXiv preprint arXiv:2505.06708 (2025).
> >
> > [8] Veličković, Petar, et al. "softmax is not enough (for sharp out-of-distribution)." arXiv preprint arXiv:2410.01104 (2024).

---

> > > ### Comment · Reviewer_Z1TA · 2025-06-03
> > >
> > > Thanks for your responses! Most of my concerns are well addressed. I kept my score but raised my confidence to 4.

---

> > > > ### Author Response · Authors · 2025-06-03
> > > > **Thank you!**
> > > >
> > > > We thank you very much once again for your efforts in the review and for supporting our paper!
> > > >
> > > > Best,
> > > > Authors

---

### Official Review · Reviewer_4hVJ · 2025-05-12

**Rating:** 5
**Confidence:** 3
**Ethics Flag:** 1

**Summary:**

This paper discusses the phenomenon of "attention sinks" in LLMs, where a significant portion of attention is consistently directed towards the first token of a sequence (often the <BOS> token). The authors propose that this behavior is not an arbitrary artifact but rather a learned mechanism that helps LLMs, particularly deep and long-context models, to avoid "over-mixing" of information, which can cause issues like rank collapse or representational collapse. The paper provides a theoretical basis for this hypothesis by connecting it to existing work on signal propagation in Transformers and extending over-squashing bounds to multi-head attention. This theoretical framework predicts that larger models and those trained on longer contexts would exhibit stronger sink behavior. Empirically, the authors validate their claims through several experiments. Perturbation analysis on Gemma 7B demonstrates that the presence of an attention sink (specifically at the <BOS> token) limits the propagation of changes in token representations across layers. Further experiments show a positive correlation between context length during pre-training and the prevalence of attention sinks, as well as an increase in sink strength with model size (demonstrated on the LLaMa 3.1 family). Finally, the paper explores the specificity of the BOS token, concluding that while explicit training with BOS influences how the sink is constructed, the tendency to form a sink at the first available token is a more general strategy to mitigate over-mixing, seems an inevitable outcome of training under such conditions.

**Reasons To Accept:**

The paper offers a possible explanation for the attention sink phenomenon, framing it as a solution to the problem of over-mixing in deep Transformers. The work connects attention sinks to established theoretical concepts like rank collapse, representational collapse, and over-squashing. The extension of over-squashing bounds to multi-head attention (Theorem 3.2) provides a useful theoretical contribution that supports their predictions. Also, empirical experiments are provided as evidence.

**Reasons To Reject:**

1. While the paper links attention sinks to preventing "over-mixing," this concept itself feels somewhat underspecified beyond its relation to existing phenomena like rank collapse, representational collapse, and over-smoothing. It's not entirely clear if "over-mixing" is a distinct, precisely defined problem that the paper newly identifies or if it's a more qualitative descriptor for the negative consequences of these known collapse phenomena. The authors state they "connect this to existing theoretical phenomena" , making the precise contribution of the "over-mixing" framing less clear.

2. Theorem 3.2 extends over-squashing bounds to multi-head attention. While a valid extension, the resulting insight—that "weighted paths across attention heads impact the sensitivity" and that sinks can reduce these path weights—feels somewhat incremental. It's not immediately obvious how this multi-head extension provides qualitatively new predictions beyond the general principle that reducing attention to other tokens (by sinking it) would reduce their influence.

3. The paper argues sinks are "useful" and sometimes feels like they are "necessary" for long-context learning. If they are indeed a necessary byproduct, further exploration of whether alternative, equally effective mechanisms to control mixing (without such attention patterns) could be learned or designed might be warranted.

---

> ### Author Response · Authors · 2025-06-01
>
> We would like to thank you very much for your thorough review. We also thank you for the excellent points you bring up, we would like to address them below.
>
> > (Q) Over-mixing is underspecified
>
> We thank you for bringing this up, as this is quite a subtle point. In this work, we use the term mixing as a qualitative mechanism, while rank collapse and representational collapse, instead as you correctly point out, refer to more precise behaviours. This is in order to be able to group together issues like rank and representational collapse that are intuitively results stemming from catastrophic mixing that results in loss of information. We have gone through the paper to make this more clear. This also nicely complements a request from Reviewer x9S1 that asked for a more detailed description of the phenomenon. We have also added a link between overmixing and the detrimental effects of loss of sharpness / increase in head entropy explored by Veličković et al. [1], which we hope will serve to further motivate the importance of this phenomenon. We hope that these changes make this aspect of our work clearer. Thank you for bringing this up!
>
> > (Q) Improved over-squashing bounds
>
> While we agree that the improved over-squashing bound is not mathematically extremely surprising, we find that it helps to provide a bit more clarity on how different attention heads tend to interact. We believe this to be important, as essentially all models now have a very large number of attention heads. In particular, we believe it gives some insight into why the attention sink occurs across attention heads throughout the layer (see for example Figure 6) as the sensitivity accumulates over all of the attention heads present in a layer. We also note that our result implicitly adds a treatment of hidden dimension, which in the original work from Barbero et al was missing. The qualitative prediction that results from this bound is that overall we expect larger models – larger hidden dimension, number of heads, number of layers – to rely on attention sinks more and more to dampen the sensitivity. We believe that this is nicely captured in Table 1, where as the models become larger, the sink metric increases, something which as far as we are aware has not been spotted before but that we believe follows nicely from our study.
>
> > (Q) Further mechanisms to control mixing
>
> We thank you for the very important question. Indeed, we believe this to be an exciting direction. In our work, we were mostly interested in understanding the phenomenon in current LLMs as we believe this could be an important opportunity for further action. We believe our work shows how attention sinks given the current softmax architectures are a clever mechanism to prevent over-mixing – thus they are a useful mechanism and not a “training bug”. Developing methods to try to mitigate the attention sink is a very clear next step, and we hope that our work provides an important stepping stone to do so in a principled manner. We also highlight that recent work [2] shows that gating (a mechanism to prevent overmixing) effectively prevents the formation of attention sinks, which validates the hypothesis that their formation is a self-protection mechanism against overmixing. Further, we are very glad to see that other works have already built upon our work to try to propose more principled solutions to the issue, but we unfortunately are not able to share them to preserve the double blind rules.
>
> We sincerely thank you for thoroughly reviewing our work. We are of course available for further discussions if needed. We kindly also point to the general comment for a list of improvements we have made to the paper.
>
>
> [1] Veličković, Petar, et al. "softmax is not enough (for sharp out-of-distribution)." ICML 2025.
> [2] Qiu, Zihan, et al. "Gated Attention for Large Language Models: Non-linearity, Sparsity, and Attention-Sink-Free." arXiv preprint arXiv:2505.06708 (2025).

---

> > ### Author Response · Authors · 2025-06-10
> >
> > Dear Reviewer 4hVJ,
> >
> > We are writing to gently follow up on our rebuttal. As the discussion period is coming to an end, we were wondering if our response has addressed your concerns and if you had any remaining thoughts? We thank you once again for your efforts and comments!
> >
> > Best,
> > Authors

---

### Official Review · Reviewer_x9S1 · 2025-05-12

**Rating:** 6
**Confidence:** 3
**Ethics Flag:** 1

**Summary:**

The paper demonstrates high technical quality through rigorous theoretical analysis and empirical validation. The authors provide proofs for key propositions (e.g., Proposition 3.1 linking rank collapse to representational collapse) and derive bounds for over-squashing in multi-head attention (Theorem 3.2). Experiments on Gemma 7B and the LLaMa 3.1 family are well-designed, with controlled variables (context length, model size) to isolate the effects of attention sinks. The perturbation analysis (Figure 2) and ablation studies (Table 2) offer compelling evidence for the proposed hypotheses. The work is methodologically sound, though some experiments (e.g., synthetic pre-training runs) could benefit from broader model diversity.

**Questions To Authors:**

See Reasons To Reject

**Reasons To Accept:**

* Novel Theoretical Contribution: The paper provides a unified framework connecting attention sinks to over-mixing, rank collapse, and representational collapse, resolving a key gap in understanding Transformer behavior.

* Empirical Rigor: Extensive experiments validate theoretical claims across multiple model families (Gemma, LLaMa) and training setups. The perturbation analysis (Figure 2) and ablation studies (Table 2) are particularly compelling.

* Practical Implications: Insights into how context length and model size influence sink formation can guide hyperparameter choices in LLM training. The analysis of data packing (Section 5) offers actionable recommendations for pre-training pipelines.

* Interdisciplinary Impact: By bridging theoretical analysis (e.g., over-squashing bounds) and empirical observations, the work enriches both the mechanistic interpretability and architecture design literatures.

**Reasons To Reject:**

* Experiments focus on LLaMa and Gemma variants. Results may not generalize to other architectures (e.g., encoder-decoder Transformers or non-autoregressive models).

* The theoretical sections (e.g., Theorem 3.2) are dense and assume familiarity with niche concepts like over-squashing. A more intuitive explanation of key terms (e.g., rank collapse) would broaden accessibility.

* While the paper explains why sinks form, it does not propose methods to mitigate potential downsides (e.g., wasted attention on <bos>). Practical utility could be strengthened with actionable mitigation strategies.

* The impact of attention sinks on task-specific performance (e.g., summarization, QA) is not explored, limiting claims about their real-world implications.

---

> ### Author Response · Authors · 2025-06-01
>
> We are very happy to read that you believe our paper “demonstrates high technical quality”, that you found our perturbation analysis and ablation study compelling, that Section 5 provides actionable recommendations for pre-training pipelines, and that this work has the potential for interdisciplinary impact.
>
> We believe you raised great points and would like to address them below.
>
> > (Q) Results may not generalise to other architectures, e.g. encoder-decoder Transformers or non-autoregressive models
>
> Our work studies decoder-Transformers as they are by far the main model used in modern LLMs today. We also note that most previous works that study attention sinks in LLMs also focus on decoder-Transformers [e.g. 1,2,3]. We are not aware of any frontier-LLM that uses either an encoder model or that is non-autoregressive (except the very recently announced Gemini-diffusion model, which has not been released yet). As such, we do not believe that this constitutes a limitation of this work. We do however believe that it is important to touch upon this, so for completeness, we have added a number of references in the background to mention that patterns similar to attention sinks have been spotted in older encoder models such as BERT as well [e.g. 4,5,6,7].
>
> > (Q) Theoretical sections are dense [...]
>
> We thank you for raising a great point, as we are very much interested in anything that could help broaden the accessibility of our work. To address this, we have added a section in the Appendix that fills the knowledge gaps on rank collapse and representational collapse. We also added a brief history of over-squashing in graph neural networks ([e.g. 8, 9]) for completeness.
>
> > (Q) Methods to mitigate potential downsides
>
> This is an important point. In this work, we focused mainly on understanding rather than mitigation because we find understanding to be a critical and valuable step.  We hope that this work can serve as a useful stepping stone for future works to propose more targeted solutions to the problem, given the ablations and results from this work. We believe great examples are for instance the work by Qiu et al [10] among many others.
>
> > (Q) Task specific performance
>
> We thank you for the suggestion. We have added an ablation with Gemma 7B on a number of real-world tasks showcasing that removing the attention sink greatly affects performance. We have also added a second Table showcasing that on the long-range Ruler benchmark (using 4k tokens), performance drops from 82.5% to 0%. We believe this helps support our claims using real-world tasks and thank you again for mentioning this.
>
> | Benchmark   | w/ <BOS> | w/o <BOS> |
> |-------------|----------|-----------|
> | ARC-e       |  80.77   |   28.49   |
> | ARC-c       |  53.50   |   22.53   |
> | PIQA        |  81.72   |   52.77   |
> | SIQA        |  48.26   |   34.70   |
> | HellaSwag   |  80.61   |   27.35   |
> | Winogrande  |  72.85   |   49.41   |
>
> | Benchmark            | w/ <BOS> | w/o <BOS> |
> |----------------------|----------|-----------|
> | Ruler 4096 Context   |  82.57   |   0.00    |
>
> We thank you very much again for your great points and for helping us improve our work. We hope that you find our responses satisfactory and that they help strengthen your opinion of our manuscript!
>
> [1] Guangxuan Xiao, et al. Efficient streaming language models with attention sinks. In The Twelfth International Conference on Learning Representations, 2024
>
> [2] Xiangming Gu, et al. When attention sink emerges in language models: An empirical view. In The Thirteenth International Conference on Learning Representations, 2025
>
> [3] Nicola Cancedda. Spectral filters, dark signals, and attention sinks. arXiv preprint
> arXiv:2402.09221, 2024.
>
> [4] Jesse Vig. A multiscale visualization of attention in the transformer model. arXiv preprint
> arXiv:1906.05714, 2019.
>
> [5] Jesse Vig and Yonatan Belinkov. Analyzing the structure of attention in a transformer
> language model. arXiv preprint arXiv:1906.04284, 2019.
>
> [6] Kevin Clark, et al, Proceedings of the 2019 ACL Workshop BlackboxNLP: Analyzing and Interpreting Neural Networks for NLP, pp. 276–286, Florence, Italy, August 2019. Association for Computational Linguistics.
>
> [7] Gino Brunner, et al. On identifiability in transformers. arXiv preprint arXiv:1908.04211, 2019.
>
> [8] Alon, Uri, and Eran Yahav. "On the bottleneck of graph neural networks and its practical implications." arXiv preprint arXiv:2006.05205 (2020).
>
> [9] Di Giovanni, Francesco, et al. "On over-squashing in message passing neural networks: The impact of width, depth, and topology." International conference on machine learning. PMLR, 2023.
>
> [10] Qiu, Zihan, et al. "Gated Attention for Large Language Models: Non-linearity, Sparsity, and Attention-Sink-Free." arXiv preprint arXiv:2505.06708 (2025).

---

> > ### Comment · Reviewer_x9S1 · 2025-06-10
> >
> > Thank you for your response, I will raise my score to 6

---

> > > ### Author Response · Authors · 2025-06-10
> > > **Thank you!**
> > >
> > > We would like to thank you again for your efforts in reviewing our paper and for your useful suggestions. We are very grateful for the score increase!
> > >
> > > Best,
> > > Authors

---

### Author Response · Authors · 2025-06-01

We are grateful to the reviewers for their comments. In particular, we thank the reviewers for their positive outlook on the novelty of our work, the strength of our empirical validations, as well as some of the practical insights offered by our work.

During the rebuttal period, we have made some improvements to our work which we would like to highlight below to help with the discussion phase:

- *Additional inference-time experiments on reasoning benchmarks:* We have run additional evaluations that check the change in performance of the Gemma 7B model on tasks like ARC-e, ARC-c, PIQA, SIQA, HellaSwag, Winogrande when removing the BoS token at inference time. These experiments confirm the detrimental effect that over-mixing has on real-world tasks and how important the BOS token is when it comes to downstream performance.
- *Additional inference-time experiments on a long-context benchmark (RULER 4096 tokens):* We carry out additional experiments on long-context reasoning benchmarks, which show that preventing over-mixing through the sink token is essential to guarantee good performance on these types of tasks. The results nicely support our theory: over-mixing worsens as the context length increases and indeed it seems like removing attention sinks in long-range tasks results in a catastrophic loss in performance.
- *More detailed future works section on mitigation strategies* We have added references to recent work that came after ours, which bypasses the appearance of attention sinks, for instance, through the use of gating mechanisms, which we believe strengthens our argument that over-mixing is detrimental in decoder Transformers. This is because a gating mechanism provides a way to “skip” the mixing effect of the attention mechanism if necessary. We will also add some comments on the implications of our findings on model efficiency alongside other references on methods that aim to mitigate attention sink effects.
- *Additional background to make the paper more accessible* We have added additional context on rank collapse, representational collapse, and over-squashing, given the suggestion from one of the reviewers to make our work accessible to a broader audience.

We would like to thank the Reviewers once again and are very glad to see that our work has been overall well received. We are very happy to answer any further questions during the remainder of the rebuttal period.

---

### Decision · Program_Chairs · 2025-07-08

**Decision:**

Accept

**Comment:**

This paper offers a novel theoretical and empirical analysis of attention sinks in LLMs, arguing they emerge as a mechanism to prevent over-mixing. The work connects attention sinks to rank and representational collapse and extends over-squashing bounds to multi-head attention. Experiments across Gemma and LLaMA models support the claims.

Pros:
- Strong theoretical grounding (Proposition 3.1, Theorem 3.2)
- Well-executed experiments with real-world benchmarks
- Clear practical implications for long-context LLM behavior

Concerns:
- "Over-mixing" was initially underspecified [4hVJ]
- Over-squashing extension seen as incremental [4hVJ]
- No direct mitigation strategy proposed [x9S1, Z1TA]
- Focused only on decoder models [x9S1]

Author Response:
The authors addressed most of concerns with new experiments, improved explanations, and additional references to mitigation strategies. Rebuttal clarified terminology, strengthened theoretical claims, and added long-context ablations.